Profiling tear proteomes of patients with unilateral relapsed Behcet’s disease-associated uveitis using data-independent acquisition proteomics

Liang Anyi 1
Qin Weiwei 2 3
Zhang Meifen 1
Gao Fei 1
Zhao Chan 1 zhaochan@pumch.cn
http://orcid.org/0000-0001-7159-2558 Gao Youhe 3 gaoyouhe@bnu.edu.cn
1 Department of Ophthalmology, Peking Union Medical College Hospital, Chinese Academy of Medical Sciences , Beijing , China
2 Department of Anesthesiology, Qingdao Municipal Hospital, Qingdao University , Qingdao , China
3 Department of Biochemistry and Molecular Biology, Gene Engineering Drug and Biotechnology Beijing Key Laboratory, Beijing Normal University , Beijing , China
Albertini Maria Cristina
Electronic publication date: 2020 Jun 19
Publication date: 2020
Volume: 8
Electronic Location ID: e9250
Received 2019 Nov 10; Accepted 2020 May 7
Copyright: © 2020 Liang et al.
Copyright year: 2020
Copyright holder: Liang et al.
License: This is an open access article distributed under the terms of the Creative Commons Attribution License, which permits unrestricted use, distribution, reproduction and adaptation in any medium and for any purpose provided that it is properly attributed. For attribution, the original author(s), title, publication source (PeerJ) and either DOI or URL of the article must be cited.
License URL: https://creativecommons.org/licenses/by/4.0/

Keywords: Behcet’s disease, Biomarkers, Intraocular inflammation, Tear proteomics, Uveitis

Funding: National Key Research and Development Program of China 2018YFC0910202 and 2016YFC1306300 Beijing Natural Science Foundation 7172076 and 7192174 Beijing cooperative construction project 110651103 Beijing Normal University 11100704 2016 PUMCH Science Fund for Junior Faculty pumch-2016-2.27 This work was supported by the National Key Research and Development Program of China (2018YFC0910202, 2016YFC1306300), Beijing Natural Science Foundation (7172076 and 7192174), Beijing cooperative construction project (110651103), Beijing Normal University (11100704) and 2016 PUMCH Science Fund for Junior Faculty (pumch-2016-2.27). The funders had no role in study design, data collection and analysis, decision to publish, or preparation of the manuscript.

==============================
Purpose

To explore whether unilateral relapse of Bechet’s disease-associated uveitis (BDU) causes differences in the tear proteome between the diseased and the contralateral quiescent eye and potential tear biomarkers for uveitis recurrence and disease monitoring.

Method

To minimize interindividual variations, bilateral tear samples were collected from the same patient (n = 15) with unilateral relapse of BDU. A data-independent acquisition (DIA) strategy was used to identify proteins that differed between active and quiescent eyes.

Results

A total of 1,797 confident proteins were identified in the tear samples, of which 381 (21.2%) were also highly expressed in various tissues and organs. Fifty-one (2.8%) proteins differed in terms of expression between tears in active and quiescent eyes, 9 (17.6%) of which were functionally related to immunity or inflammation. Alpha-1-acid glycoprotein 1 (fold change = 3.2, p = 0.007) was increased and Annexin A1 (fold change = −1.7, p < 0.001) was decreased in the tears of the active BDU eye compared to the contralateral quiescent eye.

Conclusions

A substantial amount of confident proteins were detected in the tears of BDU patients, including proteins that were deferentially expressed in the uveitis-relapsed eyes and the contralateral quiescent eyes. Some of these identified tear proteins play important roles in immune and inflammatory processes. Tear proteome might be a good source of biomarkers for uveitis.

Introduction

Tear fluid is a complex mixture of water, electrolytes, metabolites, lipids and proteins (mucins, enzymes, glycoproteins, immunoglobulins and others) secreted mainly by the main and accessory lacrimal glands (Azkargorta et al., 2017; Pieragostino et al., 2015). Tears can be easily and noninvasively accessed (Quah, Tong & Barbier, 2014) and are considered a useful source for biomarker research of ocular and systemic diseases. According to a recent review, hundreds of potential specific molecular biomarkers in tears were found to be associated with ocular diseases, including dry eye disease, keratoconus and thyroid associated orbitopathy. Other reports showed that tears can reflect the state of breast cancer, prostate cancer and multiple sclerosis (Pieragostino et al., 2015; Hagan, Martin & Enríquez-De-Salamanca, 2016; Von, Funke & Grus, 2013). Moreover, tears may reflect central metabolism in some neurological disorders (Coyle, Sibony & Johnson, 1987; Lolli & Franciotta, 2010; Lebrun et al., 2015; Calais et al., 2010).

Behcet’s disease (BD) is a chronic, relapsing systemic inflammatory disease characterized by recurrent oral aphthae, ocular involvement and a variety of other clinical features. Bechet’s disease-associated uveitis (BDU) involves about half of the BD patients and is the most common cause of morbidity (Davatchi et al., 2017; You et al., 2013). Recurrent attacks of BDU are common and accumulating damages to the fundus structures frequently leads to severe vision loss or even blindness. Unfortunately, recurrence of BDU is usually abrupt and rarely heralded by prodromes or warning signs and signs of active disease are often obscured by complications involving the refractive media such as cataract, pupillary block, etc. Therefore, there is an urgent need to discover biomarkers from easily accessible biofluids such as tears, to monitor disease activity and to predict recurrence in BDU patients.

Interestingly, despite being a symmetrical organ, the eyes of BDU patients with unilateral uveitis relapse show dramatically distinct clinical manifestations. While the quiescent eye appears completely normal, the contralateral active eye might have severe visual acuity impairment possibly caused by anterior uveitis, vitritis, retinal vasculitis, retinitis, cystoid macular edema or papillitis (Markomichelakis et al., 2011). It is possible that, considering the anatomical proximity, intraocular inflammation may cause substantial changes in the tear proteome through local mechanisms, making tears a potential resource of biomarkers for the actively inflammatory eye. To address this hypothesis and to minimize interindividual variations, tear samples were simultaneously collected from both eyes of BDU patients with unilateral uveitis relapse. Samples were preserved by our previously developed “dry” method (Qin et al., 2017) and further analyzed by a data-independent acquisition (DIA) strategy. We aimed to explore whether unilateral relapse of BDU causes differences in the tear proteome between the diseased and the contralateral quiescent eye and to identify potential tear biomarkers for uveitis recurrence and disease monitoring.

Materials and Methods

A summary of the overall experimental approach is presented in Fig.1.

Figure 1 Workflow schematic of this study.

Patients

BDU patients presenting to our center with acute unilateral relapse of panuveitis between January 2018 and June 2018 were included. The consent procedure and the study protocol were approved by the Institutional Review Board of the Institute of Basic Medical Sciences, Chinese Academy of Medical Sciences. (Project No. 007–2014). Written informed consent was obtained from each subject prior to the study.

Patients were diagnosed with BD according to the International Study Group (ISG) or International Criteria for Behcet’s Disease (ICBD) (Criteria for diagnosis of Behçet’s disease, 1990). Acute relapse of panuveitis was defined as a decrease in visual acuity and the presence of a combination of anterior uveitis (more than 0.5+ cells in the anterior chamber), vitritis (more than 1+ cells) and inflammation of the posterior segment with the presence of at least one of the following: retinal vasculitis, retinitis, cystoid macular edema and papillitis (Markomichelakis et al., 2011). The contralateral eye was reported to be quiescent for at least 2 months. Patients with any of the following conditions were excluded: (1) other ocular or orbital diseases, including allergic conjunctivitis, infectious conjunctivitis, keratitis of infectious or noninfectious cause, thyroid associated orbitopathy, primary open angle glaucoma and primary angle closure glaucoma; (2) previous ocular trauma or surgeries; (3) other forms of uveitis; (4) other systemic diseases, such as diabetes, hypertension, cardiovascular diseases, neurological disorders and irrelevant rheumatic diseases; and (5) a Schirmer test of less than 10 mm in either eye.

Tear sample collection and preservation

Tear samples were collected and preserved using a novel Schirmer strip-based dry method (Qin et al., 2017). In detail, Schirmer strips (Tianjin Jingming New Technological Devepoment Co., Ltd, Tianjin, Tianjin, China) were placed at the lateral 1/3 of the lower conjunctival sacs of both eyes for 5 min, and strips with tears exceeding 10 mm were collected. The Schirmer strips were dried immediately with a hair dryer (70 °C for 1 min) and were stored in properly labeled vacuum-sealed bags at room temperature and transferred to −80 °C freezer within 2 weeks.

Sample preparation for mass spectrometry

Tear protein extraction: The strip was cut into small pieces and transferred to a 0.6 mL tube. Next, 200 µL elution buffer (100 mM NH3HCO3, 50 mM NaCl) was added and gently shaken for 2 h at room temperature. The tube was punctured at the bottom with a cannula, placed in a larger tube (1.5 mL) and centrifuged at 12,000 g for 5 min (Posa et al., 2013). The filtrate in the outer tube was collected and quantified by the Bradford method.

Tryptic digestion: The proteins were digested with trypsin (Promega, Madison, WI, USA) using filter-aided sample preparation methods (Aass et al., 2015). Briefly, 200 µg of the protein sample was loaded on the 10-kD filter unit (Pall, NY, USA). For digestion, the protein solution was reduced with 4.5 mM DTT for 1 h at 37 °C and then alkylated with 10 mM of indoleacetic acid for 30 min at room temperature in the dark. Finally, the proteins were digested with 3 µg of trypsin for 14 h at 37 °C. The peptides were digested using Oasis HLB cartridges (Waters, Milford, MA, USA). The resulting peptides were dried and desalted in a SpeedVac (Thermo Fisher Scientific, Waltham, MA, USA). The dried peptide samples were resuspended in 0.1% formic acid and were quantified using a PierceTM BCA protein assay kit. Two micrograms of each sample were loaded for LC-MS/MS analysis using a DIA method.

High-pH fractionation: Equal volumes of tryptic digested peptides of each sample were pooled to generate the pooled sample (100 µg of peptides) for the development of a spectral library and for DIA analysis optimization and quality control. Sixty micrograms of pooled peptides was then fractionated using a high-pH spin column (Thermo Pierce, Waltham, MA, USA). After equilibration of the column, the dried sample was resuspended in 100% Buffer A (acetonitrile:H2O, 90:10 with 0.1% formic acid and 10 mM of ammonium formate) and loaded onto the column, then eluted with Buffer B at 5, 7.5, 10, 12.5, 15, 17.5, 20, 30, 50 and 70% of Buffer B (H2O with 0.1% formic acid). Fractionated samples were dried completely and resuspended in 20 μl of 0.1% formic acid. Six microliters of each of the fractions was loaded for LC-MS/MS analysis using a data-dependent acquisition (DDA) method.

LC-MS/MS setup for data-dependent and data-independent acquisition

LC-MS/MS data acquisition was performed on a Fusion Lumos mass spectrometer (Thermo Fisher Scientific, Waltham, MA, USA) interfaced with an EASY-nLC 1,000 UHPLC system (Thermo Fisher Scientific, Waltham, MA, USA). For both DDA and DIA analyses, the same LC settings were used for retention time stability. For facilitating retention time alignments among samples, a retention time kit (iRT kit from Biognosys, Schlieren, Switzerland) was spiked at a concentration of 1:20 v/v in all samples (Escher et al., 2012). The digested peptides were dissolved in 0.1% formic acid and loaded on a trap column (75 µm × 2 cm, 3 µm, C18, 100 A˚). The eluent was transferred to a reversed-phase analytical column (50 µm × 500 mm, 2 µm, C18, 100 A˚). The eluted gradient was 5–30% buffer B (0.1% formic acid in 99.9% acetonitrile; flow rate of 0.6 μl/min) for 60 min.

To acquire a spectral library for use in DIA data extraction, six μl of each of the fractions was analyzed by data-dependent acquisition. The full scan was done with a 60K resolution at 200 m/z from 350 to 1,500 m/z with an AGC target of 1e6 and a max injection time of 50 ms. Monoisotopic masses were then selected for further fragmentation for ions with a 2–6 positive charge within a dynamic exclusion range of 30 s and a minimum intensity threshold of 1e4 ions. Precursor ions were isolated using the quadrupole with an isolation window of 1.6 m/z. The most intense ions per survey scan (top speed mode) were selected for collision-induced dissociation (CID) fragmentation, and the resulting fragments were analyzed in the Orbitrap with the resolution set to 60,000. The normalized collision energy for higher energy collision dissociation (HCD)-MS2 experiments was set to 32%, the AGC target was set at 5e4 and the maximum injection time was set to 30 ms. The DDA cycle was limited to 3 s.

For DIA, 2 μg of each sample was analyzed. Survey MS scans were acquired in the Orbitrap using a range of 350–1,550 m/z with a resolution of 120,000 at 200 m/z. The AGC target was set at 1e6 with a 50 ms max injection time. Twenty-four optimal acquisition windows covered a mass range from 350 to 1,500 m/z (Table S1). The normalized collision energy for HCD-MS2 experiments was set to 32%, the AGC target was set at 2e5 and the maximum injection time was set to 54 ms. A quality control DIA analysis of the pooled sample was inserted after every ten tear samples were tested.

Data analysis

The raw MS data files acquired by the DDA mode for library construction were processed using Proteome Discoverer (version 2.1; Thermo Fisher Science, Waltham, MA, USA) with SEQUEST HT against the SwissProt human database (released in July 2016, containing 20,228 sequences) and the Biognosys iRT peptides sequences. SEQUEST HT Search parameters consisted of the parent ion mass tolerance, 10 ppm; fragment ion mass tolerance, 0.02 Da; fixed modifications, carbamidomethylated cysteine (+58.00 Da); and variable modifications, oxidized methionine (+15.995 Da). Other settings included the default parameters. All identified proteins had a false discovery rate (FDR) of ≤1%. To generate the spectral libraries, DDA spectra were analyzed as described above, and a spectral library was generated using the spectral library generation functionality of Spectronaut Pulsar (Biognosys, Schlieren, Switzerland). The library was devised by importing the search results and the raw files using Spectronaut with the default parameters.

The raw DIA files were imported to Spectronaut Pulsar, and the default settings were used for targeted analysis. In brief, a dynamic window for the XIC extraction window and a non-linear iRT calibration strategy were used. Mass calibration was set to local mass calibration. Interference correction on the MS1 and MS2 levels was enabled, removing fragments/isotopes from quantification based on the presence of interfering signals but keeping at least three for quantification. The FDR was set to 1% at the peptide precursor level and at 1% at the protein level. The significance criteria for a t-test was a p value <0.01. A minimum of two peptides matched to a protein and a fold change >1.5 were used as the criteria for the identification of differentially expressed proteins.

Results

Clinical characteristics of BDU patients

To minimize interindividual variations, bilateral tear samples were collected from the same BDU patient (n = 15) with unilateral uveitis relapse. The average age of the patients was 28.6 years old, and the female-to-male ratio was 1:4. Seven and eight patients had active inflammation in the left and right eye, respectively. The amounts of each tear samples collected by the Schirmer strips were between 10 and 30 mm (Table 1).

Table 1 Demographic characteristics, inflammatory status and amount of tears collected from each eye of the enrolled BDU patients.

Number	Sex	Age	Right eye	Left eye	
Status	Amount (mm)	Status	Amount (mm)	
1	M	23	A	20	Q	30	
2	M	30	Q	22.5	A	30	
3	F	26	Q	15	A	30	
4	M	28	Q	25	A	25	
5	M	26	A	10	Q	12.5	
6	F	32	A	12	Q	14	
7	M	18	A	25	Q	25	
8	M	28	A	15	Q	20	
9	F	29	Q	30	A	25	
10	M	28	A	10	Q	15	
11	M	22	Q	12	A	13	
12	M	34	Q	23	A	20	
13	M	25	A	17	Q	23	
14	M	33	A	27	Q	20	
15	M	48	Q	18	A	22	
Notes:

A, active (acute relapse of panuveitis); Q, quiescent (stable for at least 2 months).

There was no statistically significant difference of tear amounts between the active and the quiescent eyes (−0.27 ± 2.34 mm, p = 0.934).

Spectral library establishment and DIA method optimization

The spectral library that was generated, as described in the methods, contained 16,135 peptides corresponding to 2,779 protein groups. Prior to individual sample analysis, the pooled peptide sample was subjected to DIA experiments in order to refine the acquisition windows list and the cycle time (data points per peak). Ultimately, 25 variable windows were used (Table S1), resulting in 7–8 data points per peak (cycle time of 1.8 s). A quality control DIA analysis of the pooled sample was inserted after every 10 tear samples were tested. The number of proteins with relative abundance CV below 30% was 1,184 (85%) (Table S2), suggesting a good reproducibility of our study.

Tear proteome analysis

Tear proteome profile identified from BDU patients

With the advancement of proteomic techniques, two recent studies were able to detect more than 1,500 tear proteins (International Protein Index (IPI) protein human database version 3.76 and UniProtKB human database were used respectively) (Aass et al., 2015; Zhou et al., 2012). The comparison and number of overlapped proteins among the three studies was shown in Fig.2, and it revealed relatively high inter-study variations. In the current study, 1,797 confident proteins were identified in the 30 tear samples from both eyes of the 15 patients (Table S3). There were totally 418 (23.3%) proteins in our study that were also detected in the other two studies, with 298 (16.6%) common proteins in Zhou et al.’s (2012) study and 159 (8.8%) in Aass et al.’s (2015) study.

Figure 2 Comparison of the tear proteins identified in three studies.

We also compared the proteins detected in our study with the tissue-enriched proteome (Uhlén et al., 2015). In all, 381 (21.2%) proteins identified in the tear samples were also highly expressed in various tissues and organs. This number is comparable to the value from our previous tear proteome study which identified 365 common proteins (out of 514, 71.0%) (Qin et al., 2017).

Tear proteome variation between different disease statuses

All tear samples were assigned to either the active (A) or the quiescent (Q) group according to the inflammatory status of the eye (Table 1). Differential proteins were screened with the following criteria: fold change ≥ 1.5 between the two groups and a p-value of paired t-test < 0.05. The p-values were adjusted by Benjamini & Hochberg method. A total of 51 (2.8%) differential proteins were identified (details are described in Table S4), among which 9 (17.6%) were reported to be involved in immunity (Table 2).

Table 2 Details of the nine differential proteins related to immunity.

Uniprot ID	Protein name	Fold change*	p Value	Functions	
P02763	Alpha-1-acid glycoprotein 1	3.2	7.49E−03	Modulating the activity of the immune system during the acute-phase reaction (Ceciliani & Lecchi, 2019; Shiyan & Bovin, 1997)	
P05156	Complement factor I	1.7	4.33E−02	Responsible for cleaving the alpha-chains of C4b and C3b in the presence of the cofactors C4-binding protein and factor H respectively.	
P00734	Prothrombin	1.7	1.72E−02	Thrombin, which cleaves bonds after Arg and Lys, converts fibrinogen to fibrin and activates factors V, VII, VIII, XIII, and, in complex with thrombomodulin, protein C. Functions in blood homeostasis, inflammation and wound healing.	
P02774	Vitamin D-binding protein	1.6	1.07E−03	Enhancement of the chemotactic activity of C5 alpha for neutrophils in inflammation and macrophage activation	
P04083	Annexin A1	−1.7	2.58E−07	Play important roles in the innate immune response, anti-inflammatory (Arcone et al., 1993);
Promotes resolution of inflammation and wound healing (Leoni et al., 2015);
Promotes chemotaxis of granulocytes and monocytes (Ernst et al., 2004);
Contributes to the adaptive immune response, regulates differentiation and proliferation of activated T-cells (D’Acquisto et al., 2007);	
P10909	Clusterin	−2.1	7.06E−04	Modulate NF-kappa-B transcriptional activity (Zoubeidi et al., 2010)	
Q08380	Galectin-3-binding protein	−3.0	1.39E−03	Promotes integrin-mediated cell adhesion. May stimulate host defense against viruses and tumor cells.	
P61626	Lysozyme C	−3.5	8.47E−03	Associated with the monocyte-macrophage system, enhance the activity of immunoreagents	
P22079	Lactoperoxidase	−4.1	4.85E−04	Antimicrobial agent which utilizes hydrogen peroxide and thiocyanate (SCN) to generate the antimicrobial substance hypothiocyanous acid (HOSCN) (By similarity). May contribute to airway host defense against infection.	
Note:

* A positive fold change indicates increased protein concentration in the relapsed eye compared to the contralateral quiescent eye; a negative fold change indicates decreased protein concentration in the relapsed eye compared to the contralateral quiescent eye.

Discussion

In this study, protein profiles in the tears of BDU patients were analyzed by DIA proteomics and compared between the actively-uveitic and the contralateral quiescent eye. In total 1,797 tear proteins were detected, 51 of which were found to be differentially expressed in the inflammatory and quiet eyes. Some of these differential proteins were reported to be related to immunity or inflammation.

For comparison of tear protein lists from Zhou et al. (2012), Aass et al. (2015) and our study, we downloaded the raw data (a list of detailed protein profiles with protein name, IPI number or entry number) of the two papers and searched them in the corresponding database. Then we transformed all the proteins from the three lists to a comparable form, either protein name or entry number, to make the comparison. The inter-study variability of tear protein profiles suggests that different devices, techniques and databases might yield highly different detection results. On the other hand, the vast majorities of the tear proteins are physiologically redundant and may be influenced by a variety of genetic, environmental, local and systemic factors. While the great magnitude of variability poses challenges to tear proteomic studies, the potential of tear proteome to reflect a variety of environmental and physiological conditions also makes it a valuable resource of biomarker. The strength of our current study is that tear samples were simultaneously collected from both eyes of BDU patients with unilateral uveitis relapse, which theoretically can greatly reduce sources of variabilities including environmental, interindividual and time-dependent variations.

Noticeably, more than 300 proteins identified in our tear samples were also highly expressed in various tissues and organs. There are two possible sources of these common proteins, either constitutively expressed in eyes and excreted into tears, or come from distant organs or tissues which release these proteins into the blood circulation, and reach in tears. If it is the former, then the changes of these proteins might reflect the status of the eyes, either in physiological or pathological conditions, for example, in uveitis, and can be used to monitor ocular diseases. If the latter is true, then it further validates findings in previous studies that tear might be a good window for monitoring the change of these tissues or organs.

A total of 51 differential proteins were identified between the uveitic and contralateral quiescent eye. Notably, Alpha-1-acid glycoprotein 1 (α1-AGP), an acute phase protein produced in reaction to systemic inflammation (Fitos et al., 2006; Zsila & Iwao, 2007), was increased in active BDU. This immunomodulatory protein was found to increase in serum concentration and its glycosylation would experience qualitative changes in inflammatory status (Ceciliani & Lecchi, 2019; Shiyan & Bovin, 1997). On the other hand, Annexin A1 (ANXA1) (Arcone et al., 1993; Leoni et al., 2015; Ernst et al., 2004; D’Acquisto et al., 2007), a protein found to be anti-inflammatory, was expressed less in the tears of the active uveitic eye.

Interestingly, two immunity related proteins detected to change in our study were also reported to decrease or increase in dry eye disease (DE) (Ragland & Criss, 2017), namely ANXA1 and Lysozyme (LYZ). Noticeably, while LYZ decreased in both DE and active BDU, ANXA1 changed in the opposite directions in these two diseases. Interestingly, although LYZ was formerly known to play a role in driving a pro-inflammatory response, it is also found to have inflammation limiting and immune-dampening effects (Willcox et al., 2017). So it’s not surprising that LYZ was reduced in both active BDU and DE. Contradictorily, ANXA1 increased in DE while decreased in active BDU. ANXA1 plays multifunctional roles in innate and adaptive immunity, and is well-known for its inflammation resolution property. The decrease of ANXA1 in tears from BDU possibly implies loss of inflammation homeostasis and an actively-inflammed status in active uveitis (Bruschi et al., 2018; Gobbetti & Cooray, 2016; Sugimoto et al., 2016). But the mechanism of increase of ANXA1 in DE needs further explorations. Therefore tear proteins such as α1-AGP and ANXA1 were promising tear biomarkers for uveitis monitoring.

Candidate proteins validation, either by immunological or MS methods, is an important step in studies of biomarkers discovery. To the best of our knowledge, MS methods, multiple reaction monitoring and parallel reaction monitoring (MRM/PRM)-based approaches, are a favorable alternative to immunoassays for quantitative measurement of proteins. They are not necessarily dependent on the use of antibodies and can therefore be rapidly and cost-efficiently developed in comparison to traditional ELISAs. These advantages led many scientists to propose that affinity-based methods for protein quantification, such as ELISA or Western blot, will be soon replaced by mass spectrometric strategies (Aebersold, Burlingame & Bradshaw, 2013; Mermelekas, Vlahou & Zoidakis, 2015; Mann, 2008). So it makes more sense to validate these changed proteins in tear samples by MRM/PRM-based proteomics strategy. However, the collection of clinical tear samples was time-consuming, so we are still working on collecting more tear samples from uveitis patients including active BDU patients, quiescent BDU patients, active Vogt-Koyanagi-Harada (VKH) patients and quiescent VKH patients and carrying out further validation experiments.

This study was a primary exploratory experiment that was limited by a relatively small sample size. Enlarging the samples and enriching the grouping strategies will yield a more clinically applicable result. As was mentioned above, another limitation of our study was that the candidate biomarkers were not validated with MS based-methods. Further validation studies are needed to ensure the accuracy and reliability of the candidate biomarkers.

Conclusion

In this study, 1,797 confident proteins were detected in the tears of BDU patients, among which 51 differential proteins were identified between the uveitis-relapsed eyes and the contralateral quiescent eyes. Some of these identified tear proteins play important roles in immune and inflammatory processes, with α1-AGP and ANXA1 being promising tear biomarkers for BDU monitoring. Tear proteomes might be a good source of biomarkers for uveitis.

Supplemental Information

Supplemental Information 1 Supplemental Tables.

Table S1 shows the optimal acquisition windows. Table S2 shows the proteins with CV below 30%. Table S3 shows all the 1797 detected proteins. Table S4 shows the 51 proteins differentially expressed between the uveitis and the contralateral quiescent eyes.

Click here for additional data file.

Additional Information and Declarations

Competing Interests

Author Contributions

Human Ethics

Data Availability

The authors declare that they have no competing interests.

Anyi Liang conceived and designed the experiments, analyzed the data, prepared figures and/or tables, authored or reviewed drafts of the paper, and approved the final draft.

Weiwei Qin conceived and designed the experiments, performed the experiments, analyzed the data, prepared figures and/or tables, authored or reviewed drafts of the paper, and approved the final draft.

Meifen Zhang conceived and designed the experiments, authored or reviewed drafts of the paper, and approved the final draft.

Fei Gao conceived and designed the experiments, authored or reviewed drafts of the paper, and approved the final draft.

Chan Zhao conceived and designed the experiments, prepared figures and/or tables, authored or reviewed drafts of the paper, and approved the final draft.

Youhe Gao conceived and designed the experiments, performed the experiments, prepared figures and/or tables, authored or reviewed drafts of the paper, and approved the final draft.

The following information was supplied relating to ethical approvals (i.e., approving body and any reference numbers):

This study was approved by the Institutional Review Board of the Institute of Basic Medical Sciences at the Chinese Academy of Medical Sciences (Project No. 007–2014).

The following information was supplied regarding data availability:

The raw measurements are available in the Supplemental Files.

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
