# Peer review of "Profiling tear proteomes of patients with unilateral relapsed Behcet’s disease-associated uveitis using data-independent acquisition proteomics"

_PeerJ, doi:10.7717/peerj.9250_

## Round 0.1 · original submission · Major Revisions

The authors need to address the concerns evidenced by the reviewers to improve the manuscript.

Reviewer 1 ·

Basic reporting

NA

Experimental design

Line 162, "Other settings included the default parameters. All identified proteins had a false
162 discovery rate (FDR) of ≤1%, calculated at the peptide level." This is extremely confusing. All proteins had FDR of 1% at peptide level. So this is actually peptide FDR not protein FDR, correct? Please make sure to filter your proteins through protein FDR in addition to the current peptide FDR as the false positive problem can amplify exponentially with subsequent biomarker discovery.

Line 221, " fold change ≥ 1.5 between the two groups and a P-value of paired T-test<0.05." Did the authors perform correction for multiple hypothesis testing? Please clarify. If not, please perform the correction and use corrected p-values.

Validity of the findings

Line 201, "Of the 3718 tear proteins identified in total, only 39 proteins were detected in all three studies (Fig.2). The great inter-study variability of tear protein profiles suggests that only a small proportion of tear proteins are constitutively expressed and necessary for physiological tear functions." Incorrect claim. Not detecting a protein by proteomics does not mean it is not expressed. In fact, at least 2779 proteins were identified by the DDA method when constructing the library. There is absolutely more proteins not detected by the authors. Please rephrase. Also please comment on the technologies used by the previous studies and what potential effects that might have on the overlaps.

Line 216, "These results suggest that tear proteomes may reflect the status of other tissues/organs and thus can be a window for disease biomarker discovery" Like the author stated previously, a lot proteins are constitutively expressed, such as fibronectin etc.. Overlap with other tissues totally makes sense as these proteins are just essential to cell survival. It does not mean they can reflect status of other tissues. In fact with only 514 proteins identified, I am assuming most of these proteins are just super abundant proteins that most of cells make. Please rephrase.

Line 230, " The trends in each protein abundance changes were inconsistent among the 15 pairs of tears, but 13 (0.7%) proteins showed the same trend in more than half of the sample pairs (Table 3)." It does not make much sense by doing such comparison. This is why you do statistical test. 13 proteins having the same trend in more than half samples do not mean anything statistically. They could just go to the other extreme in the other half of the samples and would not be counted as significantly regulated proteins. If the authors really want to show proteins with same trend, please only focus on proteins that are statistically significant.

Additional comments

This is a well-designed research and it has values for people who are studying the disease. Therefore it should be published. However, there are some flaws that need to be addressed first.

Reviewer 2 ·

Basic reporting

The authors present a study using LC-MS/MS analysis to identify tear proteins differentially deregulated in the disease eye versus quiescent eye in patients with unilateral relapse of Becher`disease associate with uveitis.

Experimental design

The work has been methodologically well conducted, and high technical and methodological detail is provided in Materials & methods section.

Validity of the findings

Although the results obtained and data presented are robust and consistent, in general the discussion of results is quite short. Further discussion is recommended.
Besides there are some inconsistences between results and discussion & conclusions. For example:

In paragraph 3.1, lines 212-.217, the authors indicate “These results suggest that tear proteomes may reflect the status of other tissues/organs and thus can be a window for disease biomarker discovery”. The tears as a source of biomarkers is obvious. It has been repeatedly reported in previous publications and this statement is not a direct result of the work here presented.

In paragraph 3.2, the authors indicate that of 62 differentially expressed proteins in (A) vs (Q), authors presented in table 2 only those related with immunity. Although other ocular diseases was an exclusion criteria in the study, most of the proteins listed in table 2 have been previously reported in tear proteomics studies of dry eye (DE). I recommend authors to review the paper “TFOS DEWS II Tear Film Report. Willcox,M et al. The Ocular Surface 15 (2017) for further discussion. For example in this study Annexin A1 (ANXA1), is reduced in active BDU in comparison with quiescent, but increased in ADDE and SSDE; lysozyme is decreased in both, active BDU and DE, etc...

The number of interindividual variations presented in Table 3 is not informative. I suggest to remove this table, and maybe replace it by a table / figure presenting the physiological processes in which the remaining 54 (or total 62) are involved.

A key point in this kind of studies is the immunological validation of candidate biomarkers. I recommend to perform a validation/verification step to confirm protein expression in novel tear samples.

Conclusions paragraph presents a summary of work more than conclusions. Which are the most relevant candidate biomarkers?

Reviewer 3 ·

Basic reporting

This manuscript entitled “Profiling of tears proteome of patients with unilateral relapsed Behcet’s disease-associated uveitis via data-independent acquisition proteomics” investigates novel marker proteins on tear fluids of Behcet’s disease-associated uveitis using DIA based proteomics approach. Anyi Liang et al., carried out the comparative proteomic analysis of 15 uveitis patients resulting identification of 1,797 proteins including 62 DEPs which are significantly altered in active eye compared to the opposite quiescent eye of the same patient. This result suggests potential valuable biomarkers of Behcet’s disease-associated uveitis in tear fluids. There are several concerns, both in the structure/organization of the manuscript and their findings.

Experimental design

1) The authors compared identified proteins from tear fluids with two previous studies (in results 3.1). It is hard to understand how authors compared protein lists. This is a very critical part of your paper and needs to be explained. It makes no sense that only 1% of proteins are expressed simultaneously in tears. In the current version, the detailed comparison method are not described. It seems to be unfair comparison. For a fair comparison, download the raw data of the two papers and search using same database and parameters.
You also need an explanation for the sentence below. As you explain, the differences of expressed tear proteins among individuals should be very large, but the difference among the 15 patient's tears used in your study is not significant. This can be very confusing to readers and may mean that tear fluid cannot be used as a biomarker source. You need to have a clear explanation for this.
“The great inter-study variability of tear protein profiles suggests that only a small proportion of tear proteins are constitutively expressed and necessary for physiological tear functions.” (in line 202)

2) Following question 1, why did you compare the proteins identified in DIA, not the proteins identified in DDA?

3) In results 3.1, identified proteins in tear fluid were compared with tissue proteome data. However, there is no explanation for the comparison criteria in your manuscript. You should explain detailed criteria such as the kind of tissue you compared. Furthermore, interpretation of your data is too weak so you should describe more detailed conclusion of this data.

4) LFQ data such as DIA, DDA must be confirmed for a subset of proteins with antibody-based assays for additional validation. LFQ will miss out on isoform abundance levels in the total proteome.

5) The reproducibility of the DIA analysis was confirmed by the DDA data of the pooled peptides in triplicates. Although this can confirm the inter-day variation of LC-MS sensitivity, it is difficult to confirm that there is no variation in the analysis of tear fluid samples. To confirm the reproducibility of the DIA assay, show the CV (%) of retention time and intensity of the spiked iRT peptide.

6) What is the difference between general glaucoma and Behcet's disease-related glaucoma? There is no interpretation of the results related to Behcet disease. Please describe in the discussion part.

7) The manuscript is rather poorly organized and each section (method and results) does not serve its purpose. Especially, 3 and 4 in methods part have to be cleaned up and combined to prevent confusion for readers. In addition, the results part should be explained in more detail.

Validity of the findings

1) The amount of tear fluid sample collected from the active eye and quiescent eye in each patient and the protein per unit volume are not given. Is there any significant variation in the opposite eye?

2) Why did the authors use HLB for desalting?
“Oasis HLB cartridges (Waters, USA). The resulting peptides were dried and desalted in a SpeedVac (Thermo Fisher Scientific, Waltham, MA).” (in line 112)  should be changed.

3) “The eluent was transferred to a reversed-phase analytical column (50 μm × 500 mm, 2 μm, C18, 100 Å). The eluted gradient was 5–30% buffer B (0.1% formic acid in 99.9% acetonitrile; flow rate of 0.6 μl/min) for 60 min.” (in line 134)
 Although the used column was 50 μm (inner meter) x 50 cm (Length), you used flow rate of 600nl/min. Perhaps the maximum pressure seems to be exceeded on easy nano-LC. You did not even explain the column temperature. Please describe it exactly.

4) Revise the wrong number with four-digit in this manuscript. 1 797 should be 1,797 in line 29. Please check all typing errors in word and number.

5) Despite your sample is tear fluid not cell or tissue samples, why did the author use the FASP method for protein digestion instead of in-solution digestion? Please explain.

6) Line 128 : EASY-nLC 1000 HPLC System  should be UHPLC, not HPLC
7) “The most intense ions per survey scan (top speed mode) were selected for collision-induced dissociation fragmentation, and the resulting fragments were analyzed in the Orbitrap with the resolution set to 60 000. The normalized collision energy for HCD-MS2 experiments was set to 145 32%, the AGC target was set at 5e4 and the maximum injection time was set to 30 ms.” (in line 142)
 You should write an exact collision type to prevent confusion. CID : Collision-induced dissociation, HCD : Higher energy collision dissociation.

8) Why did the authors use only FDR of ≤1%, calculated at the peptide level? The protein level of 1% FDR should be applied in DDA analysis.

9) Table 3 is ambiguous to understand. Change the representation or format of Table 3 to make it easier to understand.

---

## Round 0.2 · Minor Revisions

There are few concerns remaining to address.

Reviewer 1 ·

Basic reporting

NA

Experimental design

NA

Validity of the findings

NA

Additional comments

The authors addressed my comments sufficiently.

Reviewer 2 ·

Basic reporting

No comment

Experimental design

No comment

Validity of the findings

No comment

Additional comments

As a general consideration, authors should have clearly indicated the page/line of changes performed in the text in response to the questions raised each reviewer. It has not been easy for me to locate the changes indicated in the Response to editor Letter.

The authors have made great efforts to address the reviewers’ concerns and have answered properly most of the questions and considerations indicated for the original version of the manuscript.

However, the main question that remains is related to validation of candidate biomarkers. Regardless of the technical used for validation, immunoassays (gold standard) or Mass Spectrometry (MS) based-methods (as argued by authors), the point is that the candidate biomarkers are not validated in the present study, and I must be clearly indicated in the discussion section as a limitation of the study. Maybe also indicate that the validation will be further presented in another manuscript.

A minor consideration:
Next references indicated in the Response to editor Letter are not included in References of manuscript.
[1] Willcox M, Argüeso P, Georgiev GA, Holopainen JM, Laurie GW, Millar TJ, Papas EB, Rolland JP, Schmidt TA, Stahl U, Suarez T, Subbaraman LN, Uçakhan OÖ, Jones L. TFOS DEWS II Tear Film Report. Ocul Surf. 2017. 15(3): 366-403.
[6] Aebersold R, Burlingame AL, Bradshaw RA. Western blots versus selected reaction monitoring assays: time to turn the tables. Mol Cell Proteomics. 2013. 12(9): 2381-2.
[7] Mermelekas G, Vlahou A, Zoidakis J. SRM/MRM targeted proteomics as a tool for biomarker validation and absolute quantification in human urine. Expert Rev Mol Diagn. 2015. 15(11): 1441-54.

---

## Round 0.3 · accepted · Accept

The paper is now ready for publication.